# The Role of Immune Cells Driving Electropathology and Atrial Fibrillation

**DOI:** 10.3390/cells13040311

**Published:** 2024-02-08

**Authors:** Mingxin Huang, Fabries G. Huiskes, Natasja M. S. de Groot, Bianca J. J. M. Brundel

**Affiliations:** 1Department of Physiology, Amsterdam UMC, Location Vrije Universiteit, Amsterdam Cardiovascular Sciences, Heart Failure and Arrhythmias, 1081 HZ Amsterdam, The Netherlands; m.huang1@amsterdamumc.nl (M.H.); f.g.huiskes@amsterdamumc.nl (F.G.H.); 2Department of Cardiology, Erasmus Medical Center, 3015 GD Rotterdam, The Netherlands; n.m.s.degroot@erasmusmc.nl

**Keywords:** atrial fibrillation, immune cell, immune markers, electropathology

## Abstract

Atrial fibrillation (AF) is the most common progressive cardiac arrhythmia worldwide and entails serious complications including stroke and heart failure. Despite decades of clinical research, the current treatment of AF is suboptimal. This is due to a lack of knowledge on the mechanistic root causes of AF. Prevailing theories indicate a key role for molecular and structural changes in driving electrical conduction abnormalities in the atria and as such triggering AF. Emerging evidence indicates the role of the altered atrial and systemic immune landscape in driving this so-called electropathology. Immune cells and immune markers play a central role in immune remodeling by exhibiting dual facets. While the activation and recruitment of immune cells contribute to maintaining atrial stability, the excessive activation and pronounced expression of immune markers can foster AF. This review delineates shifts in cardiac composition and the distribution of immune cells in the context of cardiac health and disease, especially AF. A comprehensive exploration of the functions of diverse immune cell types in AF and other cardiac diseases is essential to unravel the intricacies of immune remodeling. Usltimately, we delve into clinical evidence showcasing immune modifications in both the atrial and systemic domains among AF patients, aiming to elucidate immune markers for therapy and diagnostics.

## 1. Introduction

The incidence and prevalence of atrial fibrillation (AF) is increasing globally. According to the Framingham Heart Study (FHS), the prevalence of AF has tripled over the past 50 years, making it the most common cardiac arrhythmia in the world [1,2]. In Europe, the incidence of AF is expected to reach 14 million by 2060 [3]. AF is a serious disease as it may significantly diminish a patient’s quality of life, given its associations with severe complications such as stroke, heart failure, cognitive impairment, and cardiac arrest [1]. AF is defined as the uncoordinated electrical activation of the atria, resulting in ineffective atrial contraction [4]. Disturbances in the structural integrity of the atrial myocardium create an environment conducive to atrial remodeling, heightening the risk of electrical remodeling such as ectopic discharges [5]. This cycle contributes to the additional disruption of the atrial structure and compromised contraction [5]. This so-called electropathology fuels AF perpetuation and progression [1]. At present, AF is addressed through minimally invasive procedures, surgical interventions, and medication. Nevertheless, more than half of the patients undergoing ablation experience early recurrence [6]. Moreover, in 85% of patients, medications fail to hinder the onset or progression of AF and may potentially elevate the risk of other forms of arrhythmia [4]. Therefore, an in-depth insight into the molecular mechanisms driving AF will aid in pinpointing causative targets and, consequently, in developing effective therapeutic approaches based on these mechanisms.

Emerging evidence shows that the immune system is also modified in AF, and immune remodeling may represent one of the key elements underlying electropathology and AF [7]. Immune remodeling in AF refers to the recruitment and activation of immune cells and changes in immune molecular components, which shape a new atrial and systemic immune environment [8,9]. Modifications of the immune milieu modulate the structure of the atrial tissue and as such drive electrical conduction in the atria and AF onset. Importantly, various heart diseases, including myocardial ischemia, infection, and inflammation, result in immune cell activation in the heart and atrial structural alterations that may trigger AF onset [10,11]. A potential mechanism of AF onset may include inflammation-induced changes in ion channel function [12]. Conversely, AF-associated pathophysiological changes in the atria may induce immune remodeling [7]. However, unlike electropathology in the atria, immune remodeling extends beyond the atria by evoking a systematic immune response [7]. Since AF is a systemic disease, the causal relationship between the systemic immune response and AF is difficult to elucidate, as the two may aggravate each other. Systemic diseases affect the immune status and may contribute to the development and maintenance of AF. For example, obesity induces immune cell infiltration into the adipose tissue, especially pro-inflammatory phenotype macrophages that may trigger AF [13]. Gene expression related to inflammation and immunity is upregulated in the blood of obese AF patients [14]. Clinical studies utilizing high-frequency or complex atrial segmentation electrograms reveal locations near areas of epicardial fat that are involved in the release of paracrine inflammatory mediators that drive AF [15,16]. The increased infiltration of CD68^+^ macrophages in the atria of spontaneously hypertensive rats leads to higher atrial arrhythmia inducibility and a longer duration of induced AF [17]. Moreover, HIV patients show a unique systemic immune status. Vincenzo et al. highlight the malignant relationship between the HIV disease severity and the persistence of AF caused by HIV-driven immune changes [18]. In these patients, the whole body immune environment affects the occurrence, severity, and progression of AF. These observations suggest that the selection of treatment strategies for AF patients should also take into account the systemic immune status.

In this review, we focus on the association between immune remodeling and AF. First, we provide a basic introduction to the composition and distribution of healthy and diseased cardiac immune cells. Second, we outline the role of different immune cells in AF and additional cardiac diseases. Finally, the potential role of immune remodeling as a druggable and diagnostic target is discussed.

## 2. Composition and Distribution of Immune Cells in a Healthy Heart

In the last decade, research findings have identified the presence of immune cells in the human and animal heart, which either reside in or infiltrate the heart tissue. The immune cells include macrophages, mast cells, monocytes, neutrophils, eosinophils, B cells and T cells, dendritic-like (DC) cells, and natural killer cells (Figure 1) [19,20,21,22,23,24,25]. In healthy adult mice, immune cells make up around 4.7% of the cardiac tissue. The majority are myeloid cells, a type of blood cell that originates in the bone marrow, complemented by other populations including B cells, T cells, and non-myeloid lymphoid immune cells [26]. Spatiotemporal organ-wide gene expression and cellular mapping of the developing human heart reveal that immune cells are observed in the heart at 6.5 weeks post-conception [27]. By characterizing the cell types of the human fetal heart, Cui et al. identified three clusters of immune cells, namely mast cells, macrophages, and T and B lymphocytes [28].

Cardiac resident macrophages, derived from myeloid lineage cells, are one of the main immune cell types present in the heart and help to regulate the immune response and promote homeostasis in the heart. Early studies have shown that macrophages from the yolk sac do not express C-C chemokine receptor type 2 (CCR2), an important marker distinguishing recruited from resident macrophages. CCR2^−^ resident macrophages are abundant in the healthy heart and persist into adulthood [29]. The majority of mice cardiac macrophages express markers such as protein tyrosine phosphatase clusters of differentiation (CD) 45, cell–cell contact protein F4/80, integrin family member CD11b, the activation marker CD64, and MER receptor tyrosine kinase (MERTK). They can be further divided into four subpopulations based on differences in the expression of major histocompatibility complex (MHC) class II molecules and lymphocyte antigen 6C (Ly6C) [29]. Dick et al. further characterized a subpopulation of CCR2^−^MHCII^−^ resident macrophages expressing any combination of T cell immunoglobulin and mucin domain containing protein 4, lymphatic vessel endothelial hyaluronan receptor 1 (LYVE1), and/or folate receptor 2, both in mice and humans, which are also known as TLF^+^ macrophages [30]. TLF^+^ and MHCII^−^ resident macrophages are derived from the yolk sac and fetal monocyte progenitors, which proliferate locally in a homeostatic state and are minimally affected by monocytes from other tissues [29,30,31]. There are also unique immune cell populations in the heart, such as CCR2^+^ monocyte-derived macrophages [30,32]. Single-nucleus RNA sequencing has revealed the transcriptional signature of complex cardiac immune cell components. This includes human cardiac resident macrophages characterized by markers like the scavenger receptor CD163, collectin subfamily member 12 (COLEC12), mannose receptor C-type 1 (MRC1), E3 ubiquitin ligase membrane-associated ring-CH-type finger 1 (MARCH1), and natural resistance-associated macrophage protein 1 (NRAMP1) [21]. These proteins help macrophages to recognize endogenous and exogenous pathogens (CD163, COLEC12, MRC1) and translocate them via endocytosis (CD163, COLEC12, MRC1, NRAMP1), followed by degradation and digestion (COLEC12, MARCH1). Other immune cells in the heart include lymphocytes and mast cells. Lymphocytes are involved in immune responses and express T cell markers [11]. Mast cells are suggested to originate from the pericardial adipose tissue and contribute to immune regulation [23]. By maintaining cardiac homeostasis via protecting against infections and promoting tissue repair, the immune system plays a vital role in supporting normal heart function. A better understanding of cardiac immune cells could provide insights into heart health and disease.

Only a few studies have characterized the immune cell composition specifically in the human atria. Litviňuková et al. analyzed single-cell and single-nucleus transcriptomes from different regions of the human heart, including the left ventricle, right ventricle, left atrium, right atrium, apex, and septum [19]. Their results showed some key differences in immune cell populations between the atria and ventricles. In the atrial tissue, immune cells accounted for 10.4% of cells, compared to only 5.3% in the ventricular tissue. This suggests that the immune system may play a more prominent role in the atria. In addition, the study identified 21 distinct populations of cardiac immune cells. Some populations, such as cardiac monocyte-derived macrophages, LYVE1-expressing macrophages, dedicator of cytokinesis 4 expressing macrophages, and antigen-presenting macrophages, were found to be unique to the heart and not present in other tissues like the skeletal muscle or kidney. Other research has provided further insights into the presence of immune cells in the atria.

In the left atria of patients with sinus rhythms, the most common immune cell types were lymphocytes, mononuclear phagocytes, dendritic cells, and neutrophils [33]. Higher densities of CD68^+^CD163^+^ macrophages were seen in the human atrioventricular (AV) node versus atrial cardiomyocytes [34]. Macrophages polarize according to environmental signals, and the two classic polarization phenotypes are M1-like macrophages and M2-like macrophages. M1 macrophages can respond to pro-inflammatory signals and subsequently function to activate immunity. In contrast, M2 macrophages can respond to anti-inflammatory signals and accordingly participate in immune suppression, tissue remodeling, and repair [35]. One study found a striking feature of human cardiac macrophage populations and identified an M2-like macrophage subpopulation, called M-S2, to be highly enriched (94.0%) in the right atrium. Although the functional distinction of this subpopulation is not discussed, expression profiles suggest macrophages to be more active in the atria versus ventricles. Specifically, macrophages in the right atrium predominantly express genes involved in inflammation, like C-X-C motif chemokine ligand (CXCL) 3 and interleukin (IL)-1β, as well as genes related to antigen presentation, such as integrin subunit alpha X and complement C5a receptor 2. Meanwhile, macrophages in the left atrium stand out by expressing genes involved in ion transportation, including sodium voltage-gated channel alpha subunit 9 and solute carrier family 22 member 16 [21], where the latter has been associated with AF (Figure 1) [36].

These findings point to potential regional differences in macrophage function within the heart. The higher expression of pro-inflammatory genes in right atrial macrophages, for example, could influence atrial-specific processes like electrical conduction and contraction and arrhythmias. Further research is needed to better understand macrophage heterogeneity and their roles in heart function. Immune cells in the atria likely play important roles in maintaining atrial function and health. For example, macrophages help to regulate inflammation and tissue repair after injury [37]. A better understanding of the regional immune differences in the heart may provide new perspectives on the mechanisms underlying atrial diseases.

## 3. Composition and Distribution of Immune Cells during AF and Cardiac Disease

Any injury to the cardiac tissue triggers a cascade of dynamic cellular responses, which are initially answered by resident immune cells. The cellular responses evolve in a coordinated manner over time, leading to the recruitment of different immune cells to the inflamed tissue [38]. Resident cardiac macrophages, expressing the markers CCR2, CD11c, and LyC6, were found to expand by in situ proliferation initially in the absence of monocyte input in a mouse stress model established by angiogenin II injection, whereas CCR2-expressing monocytes were able to contribute to the CCR2^+^ and CCR2^−^ macrophage subpopulations through recruitment and proliferation [29]. In response to injury, neonatal hearts selectively increase the number of MHC-II^lo^CCR2^−^ macrophages and do not recruit additional CCR2^+^ monocytes. However, injured adult hearts selectively recruit monocytes and MHC-II^hi^CCR2^+^ monocyte-derived macrophages [39]. Immunostaining showed rapid macrophage expansion in neonatal hearts that subsided within 5 days of injury, in contrast to the progressive infiltration of monocytes and macrophages in adult hearts after injury [39]. Monocyte involvement as immune cells at homeostasis is limited. Macrophages chimerized by monocytes were found in the heart on the sixth day after bone marrow cell depletion using an associative symbiosis mouse model, suggesting that monocytes repopulate depleted cardiac macrophages [31]. Similarly, monocyte-derived macrophages replaced resident macrophages at ischemic regions of the heart after myocardial infarction [31]. These findings indicate a variety of macrophage functions during the development of cardiac diseases.

Changes in the composition and number of immune cells have also been observed in atrial-related diseases. In comparison to sinus rhythm patients, MP/DC clusters, which could be subdivided into four monocyte subclusters, including five macrophage subclusters and two DC subclusters, by reclustering analysis, were enlarged twofold in the left atria of patients with mitral regurgitation combined with AF [33]. The study further analyzed the differentially expressed gene groups between cardiac MP/DCs in the control group and AF patients, and it found that the secreted phosphoprotein 1 (SPP1) and CCR2 genes and triggering receptor expressed on myeloid cells 2 (TREM2) and CD9 cell surface markers were increased in the MP/DCs of the left atrium in AF patients. Immunofluorescence histology was subsequently used on the left atrial appendages of 108 AF patients with mitral regurgitation and 41 sinus rhythm controls, and it was observed that atrial macrophages and monocytes originated in the greatest proportion from the CCR2^+^ subpopulation [33]. In addition, greater inflammatory cell infiltration was observed in the atria of patients with AF, and these cells included macrophages expressing CD45^+^CD68^+^, T cells expressing CD3, and B cells expressing CD20. These findings were observed in the right and left atria, the right and left appendages, and the atrial endocardium and sub-endocardium (immature macrophages) and the mid-myocardium (mature macrophages), respectively [40,41,42,43,44]. The large infiltration of inflammatory cells was also detected in the adipose tissue, most notably neutrophils [45,46]. Bioinformatics analysis suggests that macrophages, monocytes, myeloid-derived suppressor cells, dendritic cells, T cells, resting NK cells, mast cells, and neutrophils are associated with different subtypes of AF [47,48,49]. Autopsy results in patients with rheumatic heart disease (*n* = 5) showed significantly higher cell counts of myeloid DC (CD11C^+^), migratory activated DCs (CD209^+^), mature DCs (CD80^+^), T lymphocytes, and M1 (CD68^+^) and M2 macrophages (CD163^+^) in the left atrium compared with those in the non-rheumatic heart disease group (*n* = 5) [50], indicating a role for atrial immune cells in rheumatic heart disease (Figure 1).

Moreover, animal experiments provide evidence that different immune cells are involved in the development of atrial dysfunction. Sarcoplasmic reticulum calcium-ATPase2a (SERCA2a) is predominantly expressed in cardiomyocytes [51] and at higher levels in the atria than in the ventricles [52]. In mice, the expression of the SERCA2a 971–990 mutant specifically induced atrial myocarditis, which was accompanied by the infiltration of monocytes and granulocytes in all mice atria (*n* = 11) and severe inflammation in the right and left atria of more than half of the mice, with the involvement of all three layers of the myocardium (endocardium, myocardium, and epicardium) in the atria [53]. Further categorization showed that in the T cell population, CD4^+^ T cells were present more frequently than CD8^+^ T cells. In the non-T-cell population, macrophages were detected more frequently than neutrophils. In a lipopolysaccharide-simulated rat model of atrial inflammation, macrophages were seen to infiltrate predominantly within the atrial endocardium 18 h after injection, and, as time progressed, macrophage infiltration diffused throughout the atrial myocardium by 72 h [54]. In a mouse model of atrial disease combining hypertension, obesity, and mitral regurgitation, more changes in atrial myeloid cells compared to ventricular cells were observed, with elevated numbers of CCR2^+^ macrophages and monocytes (Figure 1) [33]. According to the above experimental results, the immune cell component of the atrium is more inclined to differentiate in the direction of a pro-inflammation expression pattern in several atrial disease.

**Figure 1 cells-13-00311-f001:**
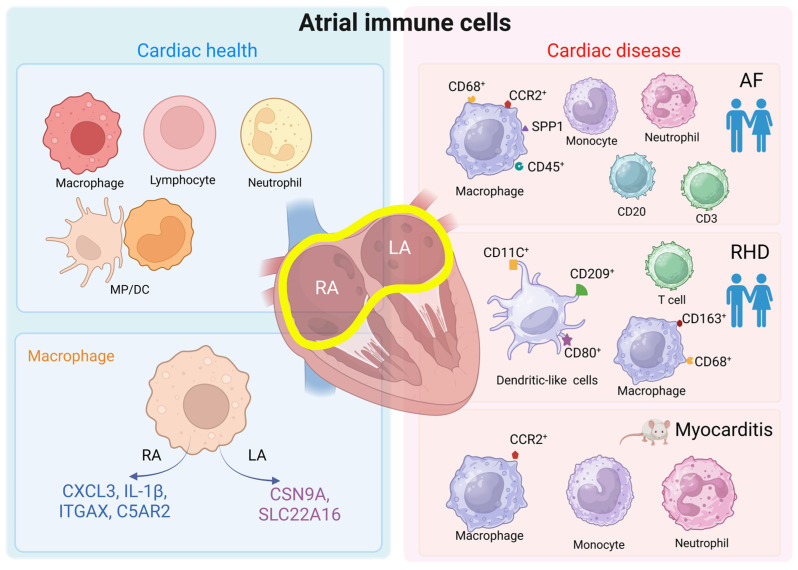
Atrial immune cells in healthy vs. cardiac disease context. The atrium in the healthy state contains most of the immune cells, such as macrophages, lymphocytes, neutrophils, monocytes, and dendritic cells. These cells are in an inactive state and have a rounded surface. During cardiac disease, atrial immune cells are activated, extend tentacles on their surface, and express different surface proteins to convey messages. We list common atrial diseases in human or animal studies concerning immune cell performance. The findings of AF and RHD are described in clinical patient cohorts, indicating clinical implications of immune cells [33,40,41,42,43,44,45,46,50]. MP/DC = mononuclear phagocytes and dendritic cells; RA = right atrium; LA = left atrium; AF = atrial fibrillation; RHD = rheumatic heart disease. Created with BioRender.com.

## 4. What Is the Function of Various Immune Cells in Cardiac Function and AF?

### 4.1. Macrophages and Monocytes

As mentioned above, cardiac macrophages are divided into two main categories based on their origin: resident macrophages of embryonic origin and macrophages differentiated from recruited peripheral blood monocytes. The individual identities of macrophages usually determine their function [55].

The functions of cardiac-resident macrophages can be divided into two categories: a classic immune function and a local tissue support function. In their study, Heidt et al. [31] observed that 11.4 ± 1.2% of cardiac macrophages exhibited fluorescence two hours after injecting fluorescent Staphylococcus aureus into the mouse heart. This finding suggests that macrophages play a role as immune fighters in immune defense. Additionally, macrophages contribute to immune stabilization. Resident macrophages support cardiac homeostasis by preventing the accumulation of waste products in the extracellular space. They achieve this by engulfing subcellular vesicles containing senescent organelles of cardiomyocytes through the recognition of the receptor MERTK and autophagic mechanisms [56]. Neonatal cardiac-resident macrophages are reparative and produce minimal inflammation or adverse remodeling after cardiac injury [39].

Cardiac-resident macrophages also have functions unrelated to immunity. Lyve^+^TLF^+^ cardiac-resident macrophages are more commonly found in the subepicardial compartment of the developing heart during cardiac development, supporting angiogenesis and lymphangiogenesis [57,58]. They are involved in driving cardiac regeneration in neonatal mice with infarction [59]. Hulsmans et al. [34] discovered that cardiac macrophages rely on surface-expressed connexin 43, which is coupled to cardiac conduction cells. This coupling modulates cardiomyocytes and facilitates electrical conduction through the distal atrioventricular (AV) node in mice. It should be noted that this result in mice is in contradiction to clinical observations, where immune-deficient patients do not show AV blockage. Macrophage ablation in mice, as demonstrated in the study, leads to AV conduction blockage. Moreover, macrophages have a negative resting membrane potential and are capable of synchronizing conduction with cardiomyocytes [34]. Although resident macrophages are depleted within the peri-infarct region, they proliferate slowly and increase in vitro, which spatially limits overall myocardial dysfunction and pathological remodeling [60].

Macrophages recruited from peripheral monocytes play important roles in both healthy and diseased states of the heart. Macrophages act as immune monitors by engulfing dying cardiomyocytes through a process called cytophagy [61]. Bone marrow-derived macrophages with the phenotype LYVE1^hi^MHC-II^lo^ are beneficial in preventing cardiac fibrosis, suggesting their involvement in the regulation of cardiac structural remodeling [62].

The adult heart becomes progressively dominated by recruited macrophages with age and exhibits a strong inflammatory response when suffering injury [39]. In response to disease or foreign pathogens, the adult heart recruits blood monocyte-derived macrophages. The increased infiltration of macrophages in the early stages of infarction promotes wound healing to reduce heart failure. In a mouse model simulating left ventricular injury with cryoinjury, infiltrating macrophages exhibit a transient but intense inflammatory response to remove cellular fragments. These fragments are replaced by inflammatory cells and granulation tissue, providing a substrate for the formation of new blood vessels (cardiovessels) and the infiltration of fibroblasts. The depletion of macrophages using chlorophosphate significantly impairs wound healing and increases remodeling and mortality after myocardial injury [63]. The functional properties of monocyte subpopulations, as investigated by Nahrendorf et al. [64], show that both Ly6C^hi^ and Ly6C^lo^ monocytes are phagocytic. However, Ly6C^hi^ monocytes express high protease activity and inflammatory factors, specializing in the breakdown of the extracellular matrix. On the other hand, Ly6C^lo^ monocytes have a reduced inflammatory function but exhibit pro-angiogenic properties. This division of labor is akin to their roles in the peripheral circulation. In order to achieve the effective resolution of inflammation, the monocyte population must transition from Ly6C^hi^ to Ly6C^lo^. If the Ly6C^hi^ population continues to expand, it can lead to hyperinflammation and impaired cardiac function [65].

AF induces the polarization of pro-inflammatory macrophages, which secrete various inflammatory factors and regulate the expression of AF-related proteins. They also modulate the activation of ion channel proteins by cross-talk with cardiomyocytes via cytokines, thereby exacerbating electrical remodeling in atrial cardiomyocytes [44,66,67]. In type 1 and 2 diabetes mellitus, macrophages and associated cytokine IL-1β increase the risk of arrhythmogenesis by prolonging the action potential duration, inducing a decrease in potassium currents in cardiomyocytes and activating ryanodine receptor 2 channels, resulting in sarcoplasmic reticulum calcium ion leakage, which is found to be a trigger for AF [68,69]. In a pressure overload rat model, endothelial cells interact with macrophages to increase heterogeneous interstitial fibrosis and prolong the atrial conduction time, making them more susceptible to AF [70]. The recruitment of monocytes to the atria via the CCR2-dependent pathway contributes to AF by releasing signals that synergistically catalyze the inflammatory response. This recruitment also stimulates fibroblast activation and induces atrial structural remodeling [33].

### 4.2. Lymphocytes: T Cells, B Cells, and Natural Killer Cells

Lymphocytes, comprising T cells, B cells, and natural killer (NK) cells, form a diverse family of immune cells in the heart, each with distinct subpopulations playing multiple roles in cardiac tissue injury and repair processes. Unlike the rapid response of neutrophils, lymphocytes reach their peak abundance on day 7 after the disruption of cardiac homeostasis [71]. The function of CD4^+^ T cells appears to be influenced by the immune environment surrounding them. In acute inflammation, CD4^+^ T cells can be activated by cardiac autoantigens, reducing the density of pro-inflammatory cell recruitment [72]. Conversely, reducing the accumulation and activation of CD4^+^ T cells in the heart during chronic inflammation has been shown to reduce collagen accumulation [73,74]. Cardioprotective T cell populations expressing CD8 and angiotensin II receptor modulate cardiac adaptive immune responses and increase the secretion of anti-inflammatory factors [75]. Regulatory T lymphocytes (Tregs) are potent suppressors of the immune response, residing in parenchymal tissues and maintaining local homeostasis. Tregs have been shown to ameliorate inflammatory cell infiltration and reduce pathological remodeling [76,77,78]. The Tregs population, which is more prevalent in female patients, secretes immunoglobulin 3 and anti-inflammatory factors to reduce the severity of inflammatory cardiomyopathy [79]. However, a phenomenon known as the transition from “physiologic Tregs” to “pathologic Tregs” has been observed, accompanied by an increase in the secretion of inflammatory cytokines, which promotes immune activation and pathological remodeling [80]. Surgical mechanical stimulation triggers CD4^+^CD28^null^ cell activation and the formation of multiple micro-scars in the myocardial tissue, which are major triggers for the development of fibrillatory impulses and reentries [81,82]. Cytotoxic lymphocytes have been found to be involved in subepicardial adipocyte death and subsequent fibrosis in AF [83].

B cells play a crucial role in adaptive immunity by secreting various cytokines and antibodies. Mature B cells secrete C-C motif chemokine ligand 7, which induces the polarization of pro-inflammatory Ly6C^hi^ monocytes, mobilizing and recruiting them to the heart [84]. Signals released from the infarcted heart activate early splenic B cells, causing them to secrete numerous cytokines that promote humoral immunity and differentiate into antibody-producing plasma cells, thereby accelerating atherosclerosis [85].

NK cells contribute to reducing myocardial collagen deposition and increasing neovascularization through specific interactions with endothelial cells [86]. The activation of NK cells also enhances the potential of cardiac progenitor cells to differentiate into cardiomyocytes [87]. Viral infections can trigger a range of adaptive immune responses. Initial lymphocytes recognize viral antigens and give rise to effector lymphocytes, leading to myocarditis [88,89].

Together, lymphocytes, including T cells, B cells, and NK cells, play diverse and critical roles in cardiac tissue injury and repair processes. The function of CD4^+^ T cells is influenced by the immune environment, with implications for acute and chronic inflammation and collagen accumulation. Cardioprotective T cell populations and regulator Tregs have significant impacts on cardiac adaptive immune responses and pathological remodeling, whereas B cells and NK cells also contribute to humoral immunity, atherosclerosis, and myocardial repair processes.

### 4.3. Neutrophils

Neutrophils, which are typically absent in a healthy heart, exhibit a rapid response to cardiac imbalances, peaking within 24 h, much earlier than inflammatory monocytes and lymphocytes [64]. Upon activation, neutrophils adhere to and migrate towards the site of injury, recruiting other cells based on chemokine concentration gradients [90]. Autocrine inflammatory factors released by neutrophils at the site of injury induce positive feedback from endothelial cells, further regulating neutrophil recruitment [91]. The interaction between neutrophils and endothelial cells triggers the respiratory burst of neutrophils, leading to oxidative damage to cardiomyocytes and the contraction of the myocardial tissue [92]. Neutrophils also phagocytose dying cells and secrete proteases that digest surrounding tissue [93,94].

In addition to their interaction with endothelial cells, neutrophils can also interact with cardiomyocytes, promoting apoptosis and activating atrial fibroblasts, which results in increased fibrous tissue [95,96]. Recent studies utilizing single-cell transcriptomics have identified different subpopulations of neutrophils at various stages of healing after myocardial infarction, all of which share a unique sialic acid binding Ig-like lectin F (SiglecFhi) signature [97]. Young neutrophils express SiglecF to a greater extent than aged neutrophils and exhibit an activated phenotype, producing reactive oxygen species and displaying a higher phagocytosis capacity [97,98,99]. Neutrophils respond rapidly but have a short lifespan, decreasing in number after 3 days and nearly disappearing after 7 days. This reduction is primarily aimed at reducing inflammation and making way for Ly6Clo macrophages [64]. Neutrophils not only engage in phagocytic activity but also play a regulatory role beyond the infiltration zone. By secreting gelatinase-associated lipid transport proteins and generating neutrophil extracellular trapping networks, neutrophils can regulate macrophage polarization and phagocytosis [100,101]. Due to their prominent pro-inflammatory role, neutrophils serve as biomarkers to predict the occurrence and severity of various cardiovascular diseases [102,103,104]. In summary, neutrophils play a pivotal role in the response to cardiac imbalances, exhibiting rapid recruitment and activation in the presence of injury. Their interactions with endothelial cells, cardiomyocytes, and other immune cells contribute to both the inflammatory response and the regulation of healing processes. Understanding the dynamics and functions of neutrophils in the context of cardiovascular diseases is crucial in developing targeted therapeutic interventions and predictive biomarkers.

### 4.4. Other Cells

Mast cells, as sentinel cells, play a crucial role in the microenvironment and responding to stimuli by detecting pathogens and recognizing damage-associated molecular patterns (DAMPs) [105]. The degranulation products of mast cells, which include chylomicrons, trypsin-like enzymes, and transforming growth factor β, act as enhancers of cardiac fibrosis and as such drive cardiac impairment [106,107,108]. Kenshi et al. found that mast cells contribute to hyperglycemia-induced AF by enhancing inflammation and fibrosis in atrial tissue [109]. However, the role of mast cells appears to be contradictory. Mast cells can secrete anti-inflammatory mediators such as IL-10, IL-13, and CXCL10 without degranulation, which act as inhibitors of cardiac fibrosis [110,111,112]. Mast cells are also involved in regulating the cardiac immune microenvironment and play a subsidiary role in the recruitment of immune cells [113]. Together, these findings indicate that mast cells serve as key players in the immune response within the cardiac microenvironment, contributing to both pro-fibrotic and anti-fibrotic processes. Understanding the intricate roles of mast cells in cardiac fibrosis and immune regulation is essential in developing targeted therapeutic strategies for cardiovascular diseases.

## 5. Immune Markers as Potential Target for AF Diagnostics and Therapy

In addition to immune cell auto-infiltration, immune-associated secretory phenotypes (IASPs) play a significant role in cardiac disease. The composition and concentration of IASPs vary in different cardiac diseases and at different stages of the disease, and they may serve as indicators of disease diagnosis and prognosis, or even as therapeutic targets. Clinical studies have explored the concentrations of classical IASPs highly associated with different immune cells in the serum or atrial tissue of patients with various atrial diseases, particularly AF (Table 1). These studies highlight the importance of immune markers as potential therapeutic targets in AF.

Oxidative stress is involved in the mechanisms of AF formation and maintenance, and the occurrence of oxidative stress is also associated with inflammation. Activated immune cells produce large amounts of reactive oxygen species (ROS), which are produced primarily by specialized enzymes, such as NADPH oxidase and myeloperoxidase (MPO), which are abundantly expressed in neutrophils, monocytes, and macrophages [114,115].

**Table 1 cells-13-00311-t001:** Overview of classic immune-associated secretory phenotypes of different immune cells in patients with different atrial diseases.

Immune Cell	Study	IASPs	Sample Source	Comparison of IASP Concentration
Monocyte/macrophage	Yamashita et al. [41]	ICAM-1, VCAM-1, MCP-1, IL-6, TGF-β	LAA	SR with a history of PAF (*n* = 5) < PeAF (*n* = 11)
	Hulsmanset al. [33]	SPP1, CCR2	LAA	SR (*n* = 41) < PeAF with or without mitral regurgitation (*n* = 82)
	Zhang et al. [116]	CXCL-1, CXCR2	Blood	SR (*n* = 31) < New-onset AF with resistant hypertension (*n* = 31)
	Wan et al. [117]	MIF	Blood	Control group (*n* = 103) < AF patients; PAF (*n* = 66) < PeAF (*n* = 68) < permanent AF (*n* = 52)
	Wang et al. [118]	Galectin-3	Blood	PAF (*n* = 162) < PeAF (*n* = 51)
	Li et al. [119]	CXCL12	Blood	Permanent AF (*n* = 68) > PAF (*n* = 74); PeAF (*n* = 128)
B cell	Matsumori et al. [120]	Immunoglobulin-free light chains, kappa and lambda	Blood	SR (*n* = 28) < lone AF (*n* = 28);
				Healthy volunteer (*n* = 28) > HF (*n* = 16)
				Kappa: controls (*n* = 75) < myocarditis (*n* = 111); lambda: controls (*n* = 75) > myocarditis (*n* = 111)
T cell	Wu et al. [121]	Th17-related cytokines	Blood	Control group (*n* = 336) < AF (*n* = 336)
Neutrophil	Meulendijks et al. [95]	MPO, NET	LA	No AF (*n* = 20) < PeAF (*n* = 14)
	Rudolph et al. [122]	MPO, elastase	RAA, Blood	No AF (*n* = 17) < PeAF (*n* = 10)
	He et al. [96]	NET	LAA, Blood	SR (*n* = 4) < AF (*n* = 4)
	Holzwirth et al. [123]	MPO	Blood	No AF (*n* = 37) < PeAF (*n* = 117)
Uncategorized immune cells	Begieneman et al. [46]	CML, VCAM-1	LAA	Control group (*n* = 9) < AF (*n* = 33)
	Rahmutula et al. [124]	Total TGF-β1 and active TGF-β1	RA	No AF (*n* = 11) < AF (*n* = 2); no AF (*n* = 11) < post-operative AF (*n* = 4)
	Zhao et al. [125]	β1-AR, pErk1/2, p38MAPK, p NF-κB	PV-MS	No AF (*n* = 12) < AF (*n* = 12)
	Kato et al. [126]	MMP2	Blood	Chronic AF (*n* = 196) > control (*n* = 873)
		IL-10		Chronic AF (*n* = 196) < control (*n* = 873)
	Yalcin et al. [127]	Anti-M2-R, anti-β1-R	Blood	Healthy control (*n* = 75) < lone AF (*n* = 75)
	Serban et al. [128]	IL-1β, IL-6, VEGF	Blood	Significantly positively correlated with the duration of atrial depolarization
	Liu et al. [129]	IL-6	Blood	In total population, >50% of AF patients (101/180) have blood IL-6 above median level

ICAM-1, intercellular cell adhesion molecule 1; VCAM-1, vascular cell adhesion molecule 1; MCP-1, monocyte chemotactic protein 1; IL, interleukin; TGF-β, transforming growth factor β; LAA, left atrial appendage; SR, sinus rhythm; PAF, paroxysmal AF; PAF, persistent AF; SPP1, secreted phosphoprotein 1; CCR2, C-C chemokine receptor type 2; CXCL, chemokine (C-X-C motif) ligand; CXCR, CXC chemokine receptor; MIF, macrophage migration inhibitory factor; Th, T helper cell; MPO, myeloperoxidase; NET, neutrophil extracellular trap; LA, left atrium; RAA, right atrial appendage; CML, Nε-(carboxymethyl)lysine; RA, right atrium; β1-AR, β1 adrenergic receptor; pErk1/2, phosphorylated extracellular-signal-regulated kinase; p38MAPK, p38 mitogen-activated protein kinase; pNF-κB, phosphorylated nuclear factor kappa B; PV-MS, pulmonary vein muscle sleeve; MMP2, matrix metalloproteinase 2; anti-M2-R, antibodies against M2-muscarinic receptors; anti-b1-R, antibodies against beta1-adrenergic receptors; VEGF, vascular endothelial growth factor.

The development and progression of heart disease do not involve only a single immune cell population, but rather stem from the coordinated efforts of multiple immune cell types, as is the case with the secretion of classical IASPs. In addition to varying concentrations in different diseases, the concentrations of IASPs are also associated with disease recurrence or clinical outcomes (Table 2). This underscores the potential of immune markers as valuable tools in understanding disease progression and guiding therapeutic interventions.

Clinical AF treatments, which target inflammation, include diuretics with anti-inflammatory effects, including ACE inhibitors and MRAs, colchicine, corticosteroids, statins, and vagus nerve stimulation (see Table 2). The main purpose of these therapies is to reduce the incidence of post-operative AF (POAF) or to limit the recurrence of AF after ablation or cardioversion therapy, by attenuating the inflammatory response to these types of surgical interventions [130,132,133,134,135,136,137,138,139,140,141,142]. The inflammatory effects of surgery are significant. Compared with pre-surgery, MPO and CRP in the blood of AF patients increased on the 1st and 2nd day after ablation surgery [143]. Even without cardiopulmonary bypass, the concentration of IL-6 in the blood increases after surgery [144].

The inflammation-attenuating effects of AF treatments are validated by the observed reductions in inflammatory markers such as CRP and IL-6 [130,135,137,138,140,141]. However, these anti-inflammatory effects do not always translate to a significant, positive AF outcome. The post-operative use of MRAs does result in reduced POAF incidence for HF patients and in decreased AF recurrence and burdens for long-standing persistent AF (PeAF) patients [133]. Post-operative colchicine treatments show inflammation-attenuating effects, as well improved AF outcomes regarding POAF and AF recurrence [132,133,134]. Similarly, the post-operative application of hydrocortisone results in a reduction in inflammation marker levels and decreased POAF incidence. A single post-operative hydrocortisone injection is not effective in reducing AF recurrence. Prednisone treatments show an inflammation-attenuating effect but only limit AF recurrence after cardioversion and show no effect on AF outcomes after ablation therapy [137,139,140]. Post-ablation therapy with atorvastatin does show a decrease in the inflammation marker CRP, but this does not result in reduced AF recurrence [141].

Inflammation-attenuating therapies also seem valuable in reducing the AF burden for PAF patients, as both ACE inhibitors or angiotensin receptor blocker therapy and vagus nerve stimulation effectively reduce inflammation marker levels and result in a decreased AF burden [131,142].

The clinical outcome of AF therapies directed at inflammation differs between studies. The treatments that are reviewed here were tested on highly variable patient populations with varying underlying (cardiac) diseases, who were undergoing multiple types of surgical interventions. This underlines the importance of patient-tailored approaches for AF treatment. Expanding the clinical toolbox for AF by considering inflammation-directed therapies creates more opportunities to find the best treatments for individual patients.

New anti-inflammatory and antioxidant therapies are explored in an increasing number of studies. The injection of IL-1β antibodies to neutralize circulating IL-1β prevents cardiac structural remodeling and reduces the incidence of AF in a mouse model of chronic kidney disease [145]. Etanercept, a TNFα inhibitor, prevents atrial arrhythmias induced after 6 weeks of swimming exercise in a mouse model and reduces atrial structural remodeling and AF vulnerability [146]. Class I histone deacetylase inhibitor reduces atrial and serum inflammatory cytokines, adipokines, and atrial immunity and adipocyte infiltration in the AF dog model, as a treatment for persistent AF [147]. Omega-3 polyunsaturated fatty acids and resveratrol and its derivatives are ideal antioxidants and may attenuate AF in experimental AF model systems [148,149,150]. The role of antioxidant vitamins needs further validation. Vitamin C may be useful in attenuating atrial remodeling, reducing ROS production in a dog model of short-term (48 h) AF [151]. However, vitamins C and E do not appear to improve atrial remodeling in a long-term (7 days) AF dog model [152].

## 6. Clinical and Future Perspectives

In this review, we highlight novel findings suggesting that phenotypic and functional modifications of various immune cells in the heart play a crucial role in the onset and progression of atrial diseases, particularly AF. Despite the lack of AF-specific biomarkers, emerging evidence from clinical trials indicates that different immune cells and immune-associated secretory phenotypes could hold potential value in AF management (Table 1). However, it is important to note that these indicators are systemic and not specific to the heart. In addition to blood samples, cardiac imaging indicators are also important, such as the left atrial strain [153,154], as imaging can provide reference evidence for the inflammatory status of the atrial tissue [155,156] and therefore may help in AF management. Furthermore, the current body of literature primarily consists of case–control and retrospective cohort studies, emphasizing the need for prospective cohort studies with larger sample sizes to establish the definitive value of these markers as predictors of clinical AF.

Recent clinical studies investigating immune markers in relation to AF particularly emphasize their association with recurrent AF and postoperative AF as primary outcomes (Table 2). However, these studies have yielded varying conclusions, suggesting that the use of systemic immune markers as target indicators for AF treatment is not yet a mature diagnostic option. Current therapeutic strategies for AF primarily aim to alleviate symptoms and complications but do not effectively target the pathogenic mechanisms driving disease progression. Therefore, exploring new strategies from an immune perspective may offer promising avenues to improve AF management.

Moving forward, future research should prioritize investigating the role of immune cells and their markers in AF, along with the underlying mechanisms involved. This approach can provide specific targets for translational medicine, enabling the development of patient-tailored immunotherapies for AF. Additionally, it is important to consider that different immune cell populations exhibit variations in immune-related blood markers, potentially leading to individual differences in AF pathogenesis. Assessing immune blood markers in specific patient groups may aid in selecting drug treatments that target the underlying immune mechanisms of AF. Therefore, considering immunotherapy for AF based on individual immune profiles and levels may reveal a clinical impact.

## Figures and Tables

**Table 2 cells-13-00311-t002:** AF treatments directed at inflammation.

Treatment		Study Type	Inflammation Markers	AF Outcome	Patient Cohort	Study
ACE inhibitors or MRAs	pre- and post-operative ramipril or spironolactone	RDBPC	CRP ↓	POAF ↔	*n* = 432, SR: valve/CABG	Pretorius et al. [130]
	ACEI or ARB therapy	clinical study	CRP ↓, IL-6 ↓	AF burden ↓	*n* = 64, PAF	Roşianu et al. [131]
	pre-operative ACEI	retrospective	n.a.	POAF ↑	*n* = 6104, CABG	Miceli et al. [132]
	post-procedural eplerenone or spirolactone	meta-analysis	n.a.	POAF ↓, AF recurrence ↓, AF burden ↓ *	*n* = 3640, HF: cardiac surgery or lsPeAF: ablation	Liu et al. [133]
Colchicine	post-operative	RDBPC substudy	n.a.	POAF ↓	*n* = 336, SR: cardiac surgery	Imazio et al. [134]
	post-procedural	RDBPC	CRP ↓, IL-6 ↓	AF recurrence ↓	*n* = 161, AF: RF ablation	Deftereos et al. [135]
	post-operative	systematic review	n.a.	POAF ↓, AF recurrence ↓	*n* = 1118, SR: pericardiotomy or PAF: ablation	Verma et al. [136]
Corticosteroids	post-operative low-dose prednisone	RBPC **	CRP ↓	AF recurrence ↓	*n* = 104, PeAF: cardioversion	Dernelis & Panaretou et al. [137]
	post-operative hydrocortisone	RDBPC	CRP ↓	POAF ↓	*n* = 241, SR: valve/CABG	Halonen et al. [138]
	post-operative single low-dose hydrocortisone injection	clinical study	n.a.	AF recurrence ↔	*n* = 209, PAF: RF ablation	Won et al. [139]
	pre- and post-operative oral prednisone	RDBPC	IL-1 ↔, IL-6 ↓, IL-8 ↓, TNF-α ↔	AF recurrence ↔	*n* = 60, PAF: ablation	Iskandar et al. [140]
Statins	post-operative atorvastatin	RDBPC	CRP ↓	AF recurrence ↔	*n* = 108, PAF and PeAF: ablation	Suleiman et al. [141]
Vagus nerve stimulation	tragus stimulation, parasym device	RDBPC	IL-6 ↓, TNF-α ↔	time spent in AF ↓	*n* = 47, PAF	Stavrakis et al. [142]

ACEI, angiotensin-converting enzyme inhibitor; ARB, angiotensin receptor blocker; MRA, mineralocorticoid receptor antagonist; R(D)BPC, randomized (double-)blinded placebo control study; CABG, coronary artery bypass graft. * Only significantly reduced for spironolactone treatment. ** Study mentions blinding with respect to therapy, but not explicit double blinding.

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
