# Peer review of "The Role of Immune Cells Driving Electropathology and Atrial Fibrillation"

_cells, 2024, doi:10.3390/cells13040311_

Round 1

Reviewer 1 Report

Comments and Suggestions for Authors

This review article describes the contribution of the immune system in electrophysiological disorders such as atrial fibrillation (AF). It covers from basic science concepts such us the role of different populations of immune cells that reside in the heart in healthy and disease conditions to clinical evidence and AF treatments targeting inflammation.

-Overall, the review addresses a relevant topic for the scientific community. However, the authors need to make the review more comprehensive and didactic. In this regard, English should be carefully reviewed through the whole document: e.g., (Line 47) Emerging evidence shows that the immune system is also modified in AF, and immune remodeling may represent one of the key elements may underlie electropathology and AF

-The authors use the electropathology term throughout the whole document. Since it is not a standard term used by the scientific community, the authors should better specify its definition.

Section 2: Composition and distribution of immune cells in a healthy heart is sometimes imprecise or vague:

-(Line 73): The macrophages derived from the yolk sac predominantly express negative C-C chemokine receptor type 2 (CCR2) required for their recruitment to injured sites, which persists into adulthood and is abundant in the healthy heart. Please note that a cell can not express a negative receptor. I would encourage to better state the receptors present in embryonically derived macrophages.

-The authors should make a better distinction of the heterogeneity and function of resident cardiac macrophages. The group of Slava Epelman has a recent review that describes three primary resident cardiac macrophage populations under healthy conditions (https://doi.org/10.1016/j.immuni.2022.08.009):

1)    Resident cardiac macrophages that renew almost exclusively through in situ proliferation. They express a combination of TIMD4, LYVE1 and FOLR2 markers (termed TLF+ macrophages by other studies)

2)    Resident cardiac macrophages that are largely replaced by monocytes and characterized by the expression of CCR2. 

3)    A third population that is partially replaced by monocytes and defined by the lack of TLF markers and CCR2 but high expression of MHC-II genes

In contrast, the authors refer to a macrophage classification from 2014 (Epelman S, Lavine KJ, Beaudin AE, Sojka DK, Carrero JA, Calderon B, et al. Embryonic and adult-derived resident cardiac macrophages are maintained through distinct mechanisms at steady state and during inflammation. Immunity. 2014 Jan;40(1):91–104). Although this is a very important reference in the field, novel technologies enabled to further refine these populations. Please include more recent and updated references.

-Are macrophage subsets present in specific locations of the heart? Can resident macrophage niches influence macrophage heterogeneity and functions?

-Surface markers may differ between species (doi: 10.3389/fphys.2022.900094). Since the review includes mouse and human information, please specify.

-Sometimes it is not clear if the authors refer to all cardiac macrophages or cardiac resident macrophages. For example, (line 83) Cardiac macrophages help regulate the immune response and promote homeostasis in the heart. They proliferate locally in a homeostatic state and are minimally affected by monocytes from other tissues. In this case they refer to resident macrophages due to the context but terminology should be more consistent through the document.

-When referring to the cellular composition of the heart the authors state that: (line 98) The majority are myeloid cells, a type o blood cell that originates in the bone marrow, with other populations including B cells, T cells, and non-myeloid lymphoid immune cells. However, previous sentences highlight that cardiac resident macrophages (abundant in homeostatic conditions) are derived from yolk salc and not from bone marrow. This can be confusing to readers.

-I find confusing to introduce the classical M1/M2 polarization in a paragraph that is referring to 2 other studies that use single cell and single nuclei RNA seq to study the diversity and heterogeneity of macrophages. M1/M2 oversimplification has been used for in vitro studies. However, this simple macrophage polarization paradigm does not adequately reflect the complex in vivo situation of the heart as demonstrated by single-cell sequencing experiments.

Section 3: Composition and distribution of immune cells during AF and cardiac disease

-(Line 154) expand by in vitro proliferation? The original publication refers to in situ proliferation. Please correct.

-(Line 158) Authors highlight the differences between macrophage populations that exits in neonatal vs adult hearts after injury. These differences have important functional implications that are not clearly stated in the text.

-(Line 171): Authors state that clusters expressing CD68 CCR2 and SPP1 can be subdivided into 4 monocytes, 5 macrophage and 2 dendritic cells clusters. Please rephrase since it may be confusing. Is the major macrophage/dentritic cluster what is further subdivided but the CD68 CCR2 and SPP1 positive cells are already a subcluster in the paper. 

-(Paragraph line 228): start with non-canonical functions of macrophages very relevant for the community but then this paragraph is mixed with canonical functions related to immunity. Parts of this are sprinkled throughout the manuscript, but I think it would be helpful to more clearly delineate the differences.

-(Line 243) frostbite refers to cryoinjury?

Comments on the Quality of English Language

Mentioned above

Author Response

Point-by-point response to the reviewers’ comments:

We thank the editor and this reviewer for their excellent suggestions and the opportunities to revise our manuscript. We have answered all the comments made by this reviewer and studied scientific literature according to the reviewers’ questions to improve our review. Changes in the revised manuscript are indicated in the manuscript and detailed point-by-point response to the comments have been included below.

Reviewer 1

Comments and Suggestions for Authors

This review article describes the contribution of the immune system in electrophysiological disorders such as atrial fibrillation (AF). It covers from basic science concepts such us the role of different populations of immune cells that reside in the heart in healthy and disease conditions to clinical evidence and AF treatments targeting inflammation.

Question 1: -Overall, the review addresses a relevant topic for the scientific community. However, the authors need to make the review more comprehensive and didactic. In this regard, English should be carefully reviewed through the whole document: e.g., (Line 47) Emerging evidence shows that the immune system is also modified in AF, and immune remodeling may represent one of the key elements may underlie electropathology and AF.

The authors use the electropathology term throughout the whole document. Since it is not a standard term used by the scientific community, the authors should better specify its definition.

Answer 1: Thank you for your suggestion. In previous articles published by our research group, electropathology was defined as ‘electrical conduction disorders, and consequently contractile dysfunction, that are caused by molecular changes in atrial tissue that drive structural changes (including myolysis, dilation and fibrosis) and AF initiation and perpetuation’ (Brundel, Nature Rev 2022, PMID: 35393446).

As suggested by this reviewer, we have added our explanation of electropathology in the manuscript (line 42).

Section 2: Composition and distribution of immune cells in a healthy heart is sometimes imprecise or vague:

Question 2: -(Line 73): The macrophages derived from the yolk sac predominantly express negative C-C chemokine receptor type 2 (CCR2) required for their recruitment to injured sites, which persists into adulthood and is abundant in the healthy heart. Please note that a cell can not express a negative receptor. I would encourage to better state the receptors present in embryonically derived macrophages.

Answer 2:  Thank you for your suggestion. C-C chemokine receptor type 2 (CCR2) is an important marker that distinguishes macrophages from recruited circulating monocytes or tissue-resident macrophages. Macrophages from the yolk sac do not express CCR2.

We have adapted the manuscript and made the correction on CCR2 macrophage expression in the manuscript, please see line 96.

Question 3: -The authors should make a better distinction of the heterogeneity and function of resident cardiac macrophages. The group of Slava Epelman has a recent review that describes three primary resident cardiac macrophage populations under healthy conditions (https://doi.org/10.1016/j.immuni.2022.08.009): 1)    Resident cardiac macrophages that renew almost exclusively through in situ proliferation. They express a combination of TIMD4, LYVE1 and FOLR2 markers (termed TLF+ macrophages by other studies) 2)    Resident cardiac macrophages that are largely replaced by monocytes and characterized by the expression of CCR2. 3)    A third population that is partially replaced by monocytes and defined by the lack of TLF markers and CCR2 but high expression of MHC-II genes.In contrast, the authors refer to a macrophage classification from 2014 (Epelman S, Lavine KJ, Beaudin AE, Sojka DK, Carrero JA, Calderon B, et al. Embryonic and adult-derived resident cardiac macrophages are maintained through distinct mechanisms at steady state and during inflammation. Immunity. 2014 Jan;40(1):91–104). Although this is a very important reference in the field, novel technologies enabled to further refine these populations. Please include more recent and updated references.

Answer 3: We sincerely appreciate your valuable comments that helped to improve the quality of our review. We have updated the literature on cardiac-resident macrophages and added additional references to cardiac-resident macrophages in the revised manuscript. We introduced different subpopulations of cardiac resident macrophages from three dimensions: life cycle, origin, and transcriptional profile characteristics. Please see section 2 paragraph 2 line 96.

Question 4: -Are macrophage subsets present in specific locations of the heart? Can resident macrophage niches influence macrophage heterogeneity and functions?

Answer 4:  Studies have shown that TLF+ resident macrophages are more abundant in the heart than in the lungs (DOI: 10.1126/sciimmunol.abf7777). Lyve+TLF+ cardiac resident macrophages are more frequently found in the subepicardial compartment of the developing heart and support vasculogenic and lymph angiogenesis (https://doi.org/10.1242/dev.194563, https://doi.org/10.1161/CIRCRESAHA.115.308270 ).

As supporting cardiac development is an important function of cardiac resident macrophages, we have added this section to the revised manuscript. See line 284.

Question 5: -Surface markers may differ between species (doi: 10.3389/fphys.2022.900094). Since the review includes mouse and human information, please specify.

Answer 5: Thank you for this comment. We have checked the literatures carefully and added in the revised manuscript information on the origin (mice or human) of the data. Please see line 85, 88, 92, 107, 115, 125, 149, 162, 168, 238,240,274,288,291,520.

Question 6: -Sometimes it is not clear if the authors refer to all cardiac macrophages or cardiac resident macrophages. For example, (line 83) Cardiac macrophages help regulate the immune response and promote homeostasis in the heart. They proliferate locally in a homeostatic state and are minimally affected by monocytes from other tissues. In this case they refer to resident macrophages due to the context but terminology should be more consistent through the document.

Answer 6: Based on your important suggestion, we now distinguish cardiac resident macrophages from cardiac recruited macrophages throughout the revised manuscript. We emphasize that the macrophages ‘are minimally affected by monocytes from other tissues’ described in line 83 (original manuscript) are cardiac resident macrophages and further clarify that these subpopulations of  macrophages refer to the TLF+ and MHCII+ macrophage subpopulations, line 117.

Question 7: -When referring to the cellular composition of the heart the authors state that: (line 98) The majority are myeloid cells, a type o blood cell that originates in the bone marrow, with other populations including B cells, T cells, and non-myeloid lymphoid immune cells. However, previous sentences highlight that cardiac resident macrophages (abundant in homeostatic conditions) are derived from yolk salc and not from bone marrow. This can be confusing to readers.

Answer 7: We were sorry that our expression may have caused confusion. We adapted the revised manuscript to clarify types of immune cells and improve readability. Please see section 2 paragraph 2 line 96.

Question 8: -I find confusing to introduce the classical M1/M2 polarization in a paragraph that is referring to 2 other studies that use single cell and single nuclei RNA seq to study the diversity and heterogeneity of macrophages. M1/M2 oversimplification has been used for in vitro studies. However, this simple macrophage polarization paradigm does not adequately reflect the complex in vivo situation of the heart as demonstrated by single-cell sequencing experiments.

Answer 8: Thank you for your question. In the single-cell RNA sequencing study of human heart tissue samples  (https://doi.org/10.1161/CIRCULATIONAHA.119.045401), as mentioned in this paragraph, the two cell clusters identified showed genetic characteristics consistent with immune cell types. One of these clusters represents cardiac resident macrophages, and subclustering of them further revealed two populations (M-S1 and M-S2) that both express M2 polarization-related genes. Subsequent comparison of differences in macrophage gene expression in different chambers revealed that the M-S2 subpopulation consisted almost exclusively of right atrial component cells. Our review mentioned the results of this study. To provide the reader with a better understanding of the subpopulation classification of macrophages, which may exhibit different functions, we provide a brief explanation of macrophage M1 and M2 polarization in the revised manuscript (line 163).

Section 3: Composition and distribution of immune cells during AF and cardiac disease

Question 9: -(Line 154) expand by in vitro proliferation? The original publication refers to in situ proliferation. Please correct.

Answer 9: We feel sorry for our carelessness. The misstatement has been corrected in our revised  manuscript. Please see line 194. Thank you for pointing out.

Question 10: -(Line 158) Authors highlight the differences between macrophage populations that exits in neonatal vs adult hearts after injury. These differences have important functional implications that are not clearly stated in the text.

Answer 10: Thank you for careful reading. In this section, we focus on changes in the subpopulation composition of macrophages in disease states. Cardiac macrophages at different ages respond differently to injury, which can lead to the differentiation of macrophages into different subpopulations. Neonates and adults are two representative age stages. Macrophage subsets in the neonatal heart are dominated by CCR2-macrophages, whereas macrophages in the adult heart are more influenced by circulating monocytes and are dominated by CCR2+macrophage. In section 4 line 271 revised manuscript, we now provide detailed information on macrophage function based on the division of cardiac macrophages into cardiac resident macrophages and macrophages differentiated by recruited peripheral blood monocytes.

Question 11: -(Line 171): Authors state that clusters expressing CD68 CCR2 and SPP1 can be subdivided into 4 monocytes, 5 macrophage and 2 dendritic cells clusters. Please rephrase since it may be confusing. Is the major macrophage/dentritic cluster what is further subdivided but the CD68 CCR2 and SPP1 positive cells are already a subcluster in the paper. 

Answer 11: We have re-written this part according to your suggestion after carefully reading the references. Please see line 211 revised manuscript.

Question 12: -(Paragraph line 228): start with non-canonical functions of macrophages very relevant for the community but then this paragraph is mixed with canonical functions related to immunity. Parts of this are sprinkled throughout the manuscript, but I think it would be helpful to more clearly delineate the differences.

Answer 12:  Thank you for your remark. Based on the previous statements about cardiac and atrial immune cells, we mainly divide cardiac immune cells into cardiac resident macrophages and monocyte-derived macrophages recruited from the periphery. In the revised manuscript, we now present the functions of cardiac macrophages according to the above classifications and further subdivide them into immune-related functions and local tissue support functions. Also we strive to make the writing more logical and easier for readers to understand. Please see section 4 line 267.

Question 13: -(Line 243) frostbite refers to cryoinjury?

Answer 13: This research used a murine cryoinjury model to induce left ventricular damage. We corrected frostbite into cryoinjury as the reference used. Please see line 312.

Reviewer 2 Report

Comments and Suggestions for Authors

The topic of atrial fibrillation (AF) and inflammation is important and timely.  The manuscript is full of useful references and information. On the other hand, the review is sometimes hard to follow and does not do as well as one might like to connect the scientific threads and propose future experiments to sort of the remaining issues.

What is clear is that by creating cardiac inflammation, you can get atrial fibrillation (e.g., cardiac surgery or sterile pericarditis models in animals).  What is not so clear is how this applies to other causes of AF such as hypertension, diabetes, and age.  I offer some suggestions of ideas to consider that might help focus this review and make it more useful.

  1. The review often wonders into topics of myocardial infarction, infection, etc. that do not have much clinical relevance to AF.  Limiting the deviations from the central theme would help.
  2. A good place to start might be with a modified table 2 and the clinical evidence that inflammation has anything to do with AF.  This table should be limited to drugs that have clear evidence of reduced inflammation. For example, one trial with statins showed no effect on AF but also showed no change in inflammation (PMID: 20946227).
  3. The authors might consider the relationship of inflammation to oxidative stress, a well-accepted cause of AF.
  4. The field of the macrophages in the heart is confusing. The authors might consider some issues:
    1. Do cell surface markers always correlate with inflammatory phenotype?
    2. In general, the cardiac field does not use the M1 and M2 terminology, preferring inflammatory and anti-inflammatory.
    3. Is there any evidence that the many subtypes of macrophages identified by cell surface markers matter, or is inflammatory and anti-inflammatory enough?
    4. What evidence is there that origin of the macrophage has any impact on its role.  Clearly, when resident macrophages are depleted, they reconstitute themselves with the presence of a yolk sac, so what is the clinical importance of the distinction about origin, especially since the cells are so plastic?
    5. There is clear evidence that macrophages can be arrhythmogenic with a defined mechanism (albeit in ventricle, PMID: 33532665), and there is data that inflammatory macrophages can cause AF (PMID: 37440641).  These data might be emphasized.
    6. The authors might discuss in which conditions (hypertension, diabetes, age) are macrophages activated and how they affect cardiomyocytes and other heart cell types to cause arrhythmia.
  5. It is possible that WBCs are influenced by cardiomyocytes.  Therefore, there may be bidirectional crosstalk.
  6. The authors might consider whether IL1 inhibition, antioxidants, or other nontraditional anti-inflammatory drugs would be desirable for AF.
  7. AF initiating factors and sustaining factors may not be the same.  Where do WBCs fit in, as initiators or as sustainers?
  8. Do other WBC cell types act on macrophages or on cardiomyocytes directly?  Are these cell types necessary for common, acquired AF?
  9. When the authors talk about AF as systemic disease, this should be discussed and supported better.  Is AF just a manifestation of a systemic disease such as diabetes, or does AF cause a systemic disease?
  10. Do WBC cell type distributions vary significantly from blood, or are cardiac WBCs passively distributed from blood?  Is there evidence that cardiac WBCs differ significant in function and clinical implication from any other tissue WBCs?
  11. Does pericardial inflammation have the same effect as cardiac inflammation?
  12. In the figure, the authors should provide references and clinical implications of morphological changes of WBCs in cardiac disease.
  13. The quoted study about macrophages and AV conduction likely has no clinical implication, since adult humans with WBC depletion (such as in chemotherapy) do not get AV block.
  14. The paragraph on lymphocytes is too long and should be broken into multiple paragraphs with clear topic sentences and supporting information.
  15. The authors might review what is known about anti-inflammatory therapies such as IL1beta inhibition or special proresolving molecules and AF.
  16. Before discussing the SERCA model, more context about the model and its relevance is needed.

Author Response

Point-by-point response to the reviewers’ comments:

We thank the editor and this reviewer for their excellent suggestions and the opportunities to revise our manuscript. We have answered all the comments made by this reviewer and studied scientific literature according to the reviewers’ questions to improve our review. Changes in the revised manuscript are indicated in the manuscript and detailed point-by-point response to the comments have been included below.

Reviewer 2

Comments and Suggestions for Authors

The topic of atrial fibrillation (AF) and inflammation is important and timely.  The manuscript is full of useful references and information. On the other hand, the review is sometimes hard to follow and does not do as well as one might like to connect the scientific threads and propose future experiments to sort of the remaining issues.

What is clear is that by creating cardiac inflammation, you can get atrial fibrillation (e.g., cardiac surgery or sterile pericarditis models in animals).  What is not so clear is how this applies to other causes of AF such as hypertension, diabetes, and age.  I offer some suggestions of ideas to consider that might help focus this review and make it more useful.

  • The review often wonders into topics of myocardial infarction, infection, etc. that do not have much clinical relevance to AF.  Limiting the deviations from the central theme would help.

Answer 1: Thank you for your suggestion. Many cardiac diseases including myocardial infarction are actualy risk factor for AF. In addition, infection may result in cardiac immune cell activation and cause  inflammation that drives cardiac structural substrate alterations and AF. We have added the role of myocardial infarction, infection as risk factors for AF in line 55 and inflammation as trigger for AF in line 58 in the revised manuscript.

  • A good place to start might be with a modified table 2 and the clinical evidence that inflammation has anything to do with AF.  This table should be limited to drugs that have clear evidence of reduced inflammation. For example, one trial with statins showed no effect on AF but also showed no change in inflammation (PMID: 20946227).

Answer 2: Thank you for this suggestion. The scope of table 2 is to provide an overview of all clinical studies that are targeting inflammation and AF. These studies, independent of their results, indicate that there is clinical interest towards the role of inflammation in AF and highlights the discrepancy in efficacy to attenuate AF. Unfortunately, no clinical trial results are available with drugs that have a clear beneficial effect on inflammation and AF. As the current information in table 2 provides an overview on insights into the clinical role of specific immune mechanisms in AF, we believe it is relevant to the reader.

We adapted the text and emphasize the variation in AF clinical outcomes with drugs directed at inflammation. We also mention that differences in individual substrate may be a reason for variation in outcome and that a more patient-tailored approach is required (line 510).

  • The authors might consider the relationship of inflammation to oxidative stress, a well-accepted cause of AF.

Answer 3: Thank you for your suggestion. Both inflammation and oxidative stress are indeed important mechanisms for AF onset and maintenance and is of interest to discuss. In the revised manuscript, we added evidence to support that activated immune cells are an important source for oxidative stress. Please see line 449.  

  • The field of the macrophages in the heart is confusing. The authors might consider some issues:
    1. Do cell surface markers always correlate with inflammatory phenotype?

Answer 4.a: Research findings indicate that the inflammatory phenotype or function of immune cells is mainly distinguished by cell surface markers (PMID: 31732166, PMID: 32553181, PMID: 25035951). For example, CCR2+ macrophages tend to represent a pro-inflammatory phenotype, while CCR2- macrophages tend to support the local tissue. In the revised manuscript, we added this information in section 4 line 271.

  1. In general, the cardiac field does not use the M1 and M2 terminology, preferring inflammatory and anti-inflammatory.

Answer 4.b: Because  the cited literature in the manuscript utilize M2 in their studies (PMID: 32403949), we adapted the same expression in our manuscript. To make it easier for readers to understand, we give a brief introduction to these two phenotypes in line 165.

  1. Is there any evidence that the many subtypes of macrophages identified by cell surface markers matter, or is inflammatory and anti-inflammatory enough?

Answer 4.c: Different subpopulations of macrophages indeed are distinguished according to Macrophage cell surface markers. Different cell markers are also relevant to different functions.  Inflammatory and anti-inflammatory both are important function for macrophages, but are not enough.  In the revised manuscript, we present the cardiac macrophages according to the cardiac resident macrophages and monocyte-derived macrophages recruited from the periphery, and further discuss  their functions by subdividing them into immune-related functions and local tissue support functions, not just only from inflammatory and anti-inflammatory (see section 4 line 267).

  1. What evidence is there that origin of the macrophage has any impact on its role.  Clearly, when resident macrophages are depleted, they reconstitute themselves with the presence of a yolk sac, so what is the clinical importance of the distinction about origin, especially since the cells are so plastic?

Answer 4.d: The origins of resident macrophages include yolk sac, fetal monocyte progenitors, and circulating monocytes. With age or disaeses, a proportion of resident macrophages are replaced by recruited macrophages (line 206, 313). Macrophages come from different sources, have different markers on their surfaces, and play different functions. We can understand the immune status of the heart at the current stage through its surface markers and then provide targeted treatment.

  1. There is clear evidence that macrophages can be arrhythmogenic with a defined mechanism (albeit in ventricle, PMID: 33532665), and there is data that inflammatory macrophages can cause AF (PMID: 37440641).  These data might be emphasized.

Answer 4.e: Thank you for your supplement. Macrophages (PMID: 27882934) and its associated cytokine IL-1β (PMID:33532665) increase the risk of arrhythmogenesis in type 1 and type 2 diabetes. Underlying mechanisms include 1) prolongation of action potential duration, 2) inducing a decrease in potassium currents in cardiomyocytes, and 3) activating ryanodine receptor 2 channels causing calcium leakage from the sarcoplasmic reticulum. We have added this information in the new revised manuscript. Please see line 330.

  1. The authors might discuss in which conditions (hypertension, diabetes, age) are macrophages activated and how they affect cardiomyocytes and other heart cell types to cause arrhythmia.

Answer 4.f: In different disease contexts, macrophages may respond differently to the occurrence of AF. In the new revised manuscript, we have incorporated your suggestions and added the effects of macrophages on cardiomyocytes to cause AF in type 1 diabetes, type 2 diabetes, hypertension, etc. See line 330.

  • It is possible that WBCs are influenced by cardiomyocytes.  Therefore, there may be bidirectional crosstalk.

Answer 5: Bi-directional crosstalk between cardiomyocytes and resident immune cells is definitely an important consideration in cardiac immune remodeling and worth discussing in our review paper. With the advent of snRNA-seq comes the possibility of evaluating bi-directional cell-cell communication. Unfortunately, current research mainly reports on cell type distribution, as discussed in our review, and does not go into cell type specific ligand-receptor interactions. An in vitro study of HL-1 cardiomyocytes and macrophages in co-culture did uncover a macrophage-cardiomyocyte interaction that is induced by AF. We described these findings in our revised manuscript section 4.1 (line 328).

  • The authors might consider whether IL1 inhibition, antioxidants, or other nontraditional anti-inflammatory drugs would be desirable for AF.

Answer 6: Thank you for your reminder. An increasing number of studies have attempted to explore the possibility of anti-inflammatory and antioxidant therapy for AF, based on the findings from animal studies. As you suggested, we now added these therapies to our revised manuscript. Please see line 516.

  • AF initiating factors and sustaining factors may not be the same.  Where do WBCs fit in, as initiators or as sustainers?

Answer 7:  Thank you for your question. We believe different phenotypes of immune cells and their relevant secreted cytokines reflect different stages of AF. For example, the active macrophages (SPP1+CCR2+) could promote the onset of AF (PMID: 37440641), see line 336, but active macrophages in the  left atrial appendage are more frequently found in paroxysmal AF patients compared to persistent AF (PMID: 20009387), see table 1 line 2.  However, the mechanism how immune cells result in electrical  and structural remodeling is unclear and that is why we need more researches to uncover it.

  • Do other WBC cell types act on macrophages or on cardiomyocytes directly?  Are these cell types necessary for common, acquired AF?

Answer 8: Macrophages, and other immune cells may result in AF. For example, the toxic effect of T cells cause the formation of tiny scars on the atrium, promoting the occurrence of postoperative AF (PMID: 34536081, PMID: 32594507). We added this information in the revised manuscript in line 360.

  • When the authors talk about AF as systemic disease, this should be discussed and supported better.  Is AF just a manifestation of a systemic disease such as diabetes, or does AF cause a systemic disease?

Answer 9: In addition to cardiac diseases and injuries, systemic diseases affect the immune status and may contribute to the development of atrial fibrillation. We have therefore added this section to the revised manuscript. Please see line 64, 330.

  • Do WBC cell type distributions vary significantly from blood, or are cardiac WBCs passively distributed from blood?  Is there evidence that cardiac WBCs differ significant in function and clinical implication from any other tissue WBCs?

Answer 10: It is interesting to consider circulating immune cell populations as a reflection of resident cardiac immune cells, and therefore they may act as a biomarker. Circulating and resident immune populations are distinct, as discussed in section 2. As discussed in the response to question 4 by reviewer 1, resident cardiac macrophage differ from pulmonary resident macrophages. We could not find any studies that directly compared circulating immune cell populations to resident tissue immune cell distributions.

  • Does pericardial inflammation have the same effect as cardiac inflammation?

Answer 11: Thank you for your question. We believe pericardial inflammation also plays an important role in AF, especially for patients with obesity. Adipose tissue would call immune cells infiltration, most of these are pro-inflammatory. Mast cells originate from the pericardial adipose tissue. These immune cells are reported to be related to AF. We mention this topic in line 65, 134, 229 in the revised manuscript

  • In the figure, the authors should provide references and clinical implications of morphological changes of WBCs in cardiac disease.

Answer 12: Thank you for your suggestion. To keep the figure easy to understand, we do not include reference within the figure but all references can be found in the manuscript and the figure legend (section 3). In this figure, we mentioned the findings of AF and RHD were from clinical patients cohort, which indicated the clinical implication of immune cells . We adapted the legend of the figure to emphasize ‘clinical implications’.

  • The quoted study about macrophages and AV conduction likely has no clinical implication, since adult humans with WBC depletion (such as in chemotherapy) do not get AV block.

Answer 13: Thank you for the insight. This is an interesting point of discussion towards the direct clinical relevance of the study in question. Also the observation is still very much relevant to illustrate a possible role of macrophages in cardiac conduction however, and we have rephrased our description of the study to emphasize this (see line 290)

  • The paragraph on lymphocytes is too long and should be broken into multiple paragraphs with clear topic sentences and supporting information.

Answer 14Thank you for your suggestion. We have rewritten the paragraph on lymphocyte function into a segmented description based on different types of lymphocytes (T cell, B cell and NK cell).

  • The authors might review what is known about anti-inflammatory therapies such as IL1beta inhibition or special proresolving molecules and AF.

Answer 15: Thank you for your reminder. An increasing number of studies have attempted to explore the possibility of Anti-inflammatory and antioxidant therapy for AF, based on the findings from animal studies. As you suggested, we added these clinical studies in our manuscript (see 516).

  • Before discussing the SERCA model, more context about the model and its relevance is needed.

Answer 16: Thank you for your suggestion. We have added the background information about Sarcoplasmic/endoplasmic reticulum calcium-ATPase2a (SERCA2a) in our revised manuscript. Please see line 235.

Reviewer 3 Report

Comments and Suggestions for Authors

Congratulations to the authors for the very interesting idea of the manuscript: our knowledge on AF mechanisms isn’t growing as fast as its worldwide burden; a comprehensive understanding about one the main pathogenetic driver of this arrhythmia is indeed useful.

The review is well conducted, with clearly expresses concepts and it explores every aspects of the topic; however I have some comments in order to make this review more complete:

-        Didn’t you find any paper with correlation between immunological changes and atrial strain?

-        I think you should consider this paper: “La Fazia VM, Pierucci N, Mohanty S, Gianni C, Della Rocca DG, Compagnucci P, MacDonald B, Mayedo A, Torlapati PG, Bassiouny M, Gallinghouse GJ, Burkhardt JD, Horton R, Al-Ahmad A, Di Biase L, Natale A. Catheter ablation approach and outcome in HIV+ patients with recurrent atrial fibrillation. J Cardiovasc Electrophysiol. 2023 Dec;34(12):2527-2534. “in which the authors describe accurately  the vicious relationship between HIV disease severity, atrial cardiomyopathy caused by immunological alterations driven by HIV, left atrial enlargement and AF persistence; and taking this into account, catheter ablation of AF emerges as an effective tool to treat consequences of altered immune system atrial cardiomyopathy, ensuring sinus rhythm also in this category of patients.

-        Do any studies regarding the different immunological profiles between cardiac surgery AF and non cardiac surgery AF exist? Because Cardiac surgery due to the wide use of extra corporeal circulation determines an high activation of the innate component of immune system.

-        Does it exist any study that correlates cardiac magnetic resonance findings with altered immunological consequences on left atrial strain?

Author Response

Point-by-point response to the reviewers’ comments:

We thank the editor and this reviewer for their excellent suggestions and the opportunities to revise our manuscript. We have answered all the comments made by this reviewer and studied scientific literature according to the reviewers’ questions to improve our review. Changes in the revised manuscript are indicated in the manuscript and detailed point-by-point response to the comments have been included below.

Reviewer 3

Comments and Suggestions for Authors

Congratulations to the authors for the very interesting idea of the manuscript: our knowledge on AF mechanisms isn’t growing as fast as its worldwide burden; a comprehensive understanding about one the main pathogenetic driver of this arrhythmia is indeed useful.

The review is well conducted, with clearly expresses concepts and it explores every aspects of the topic; however I have some comments in order to make this review more complete:

Question 1:  - Didn’t you find any paper with correlation between immunological changes and atrial strain?

Answer 1: Thanks for your question. Atrial strain is an important cardiac imaging indicator. Left atrial strain is a recommended measurement in AF imaging applications to assess left atrial remodeling, thromboembolic risk, and prognosis and management in patients with atrial fibrillation (DOI: 10.1093/ehjci/jev354; DOI: 10.1016/j.echo.2014.03.010). One study showed that in the elderly, reduced left atrial strain was associated with a set of circulating inflammatory markers, Gal-3 and MMP-9 (DOI: 10.1159/000522632). In clinical study in the context of COVID-19, left atrial strain holds promise as an indicator of identifying patients at risk for a long-term inflammatory state (https://doi.org/10.1186/s13089-022-00302-5 ).

We added info on atrial strain to the section ‘future directions in research’ line 534.

Question 2:  - I think you should consider this paper: “La Fazia VM, Pierucci N, Mohanty S, Gianni C, Della Rocca DG, Compagnucci P, MacDonald B, Mayedo A, Torlapati PG, Bassiouny M, Gallinghouse GJ, Burkhardt JD, Horton R, Al-Ahmad A, Di Biase L, Natale A. Catheter ablation approach and outcome in HIV+ patients with recurrent atrial fibrillation. J Cardiovasc Electrophysiol. 2023 Dec;34(12):2527-2534. “in which the authors describe accurately  the vicious relationship between HIV disease severity, atrial cardiomyopathy caused by immunological alterations driven by HIV, left atrial enlargement and AF persistence; and taking this into account, catheter ablation of AF emerges as an effective tool to treat consequences of altered immune system atrial cardiomyopathy, ensuring sinus rhythm also in this category of patients.

Answer 2: Thank you for your recommendation. The mentioned article is of interest and provides evidence to prove AF is a systemic disease. HIV patients own the special systemic immune status. In the article, the authors highlight the malignant relationship between HIV disease severity and persistence of AF caused by HIV-driven immune changes. The immune environment not only inner atrium but also the whole body could affects the occurrence, severity, and progression of AF. Therefore, the selection of treatment methods for AF patients should also take into account the systemic immune status.

We adapted the manuscript accordingly, see line 71.

Question 3:  - Do any studies regarding the different immunological profiles between cardiac surgery AF and non cardiac surgery AF exist? Because Cardiac surgery due to the wide use of extra corporeal circulation determines an high activation of the innate component of immune system.

Answer 3: Thank you for your question. Extracorporeal circulation and surgical trauma can lead to the production of different pro-inflammatory mediators, along with widespread endothelial activation, leading to increased expression of adhesion molecules and impaired nitric oxide release. Elevated levels of IL-6, TNF-α, and CRP are all associated with POAF (please see table 2). But, eliminating the use of cardiopulmonary bypass (off-pump) may not completely reduce the incidence of POAF (https: //doi.org/10.1016/j.ejcts.2005.12.028 ). This is interesting topic we need to consider when studying the effect of surgery on recurrence of atrial fibrillation or postoperative AF. We adapt the manuscript, see line 487.

Question 4:  - Does it exist any study that correlates cardiac magnetic resonance findings with altered immunological consequences on left atrial strain?

Answer 4: This article used cardiac magnetic resonance to detect indicators of atrial strain and found that the elderly, reduced left atrial strain was associated with a set of circulating inflammatory markers, Gal-3 and MMP-9 (DOI: 10.1159/000522632). Just as the answer 1, we believe the relationship between imaging parameter such as left atrial strain and inflammatory status is potential topic for future research of AF. So we added it on the section 6 line 534.

Round 2

Reviewer 3 Report

Comments and Suggestions for Authors

Thanks for reading and appreaciating my comments. Now I think that your paper is suitable for publication

Author Response

Point-by-point response to the reviewer’s comments:

Reviewer 3

Comments and Suggestions for Authors

Comment 1: Thanks for reading and appreciating my comments. Now I think that your paper is suitable for publication

Answer 1:  We appreciate your useful comments and suggestions to improve our review and like to thank you for your positive remark.